# Clinical Proteomics in Colorectal Cancer, a Promising Tool for Improving Personalised Medicine

**DOI:** 10.3390/proteomes6040049

**Published:** 2018-12-02

**Authors:** Anaïs Chauvin, François-Michel Boisvert

**Affiliations:** Department of Anatomy and Cell Biology, Université de Sherbrooke, 3201 Jean-Mignault, Sherbrooke, QC J1E 4K8, Canada; Anais.Chauvin@USherbrooke.ca

**Keywords:** colorectal cancer, clinical proteomics, biomarkers, predictive biomarkers, personalised medicine

## Abstract

Colorectal cancer is the third most common and the fourth most lethal cancer worldwide. In most of cases, patients are diagnosed at an advanced or even metastatic stage, thus explaining the high mortality. The lack of proper clinical tests and the complicated procedures currently used for detecting this cancer, as well as for predicting the response to treatment and the outcome of a patient’s resistance in guiding clinical practice, are key elements driving the search for biomarkers. In the present overview, the different biomarkers (diagnostic, prognostic, treatment resistance) discovered through proteomics studies in various colorectal cancer study models (blood, stool, biopsies), including the different proteomic techniques used for the discovery of these biomarkers, are reviewed, as well as the various tests used in clinical practice and those currently in clinical phase. These studies define the limits and perspectives related to proteomic biomarker research for personalised medicine in colorectal cancer.

## 1. Introduction

### 1.1. Colorectal Cancer

According to the International Agency for Research on Cancer (IARC), colorectal cancer (CRC) currently represents the third most common cancer worldwide (10.2% of new cancer cases), behind lung cancer and breast cancer (each 11.6%), and the second most deadly cancer worldwide (9.2% of cancer-related deaths), behind lung cancer (18.4%), with approximately 1,900,000 new cases of CRC and 900,000 CRC-related deaths each year [1]. With regard to the five-year survival rate, the earlier the cancer is diagnosed, the higher the survival rate. Accordingly, when the cancer is diagnosed at a localised stage (stages 0 and I), the five-year survival rate approaches 90%, while decreasing to around 10% for stage IV (distant metastases) [2]. This strong decrease shows the importance of early detection of CRC; however, in most instances, CRC is diagnosed late due to the absence of symptoms and typically reveals advanced or even metastatic stage tumours, hence explaining the high mortality rate. All of these statistical data demonstrate the importance of biomarker research, particularly for the early diagnosis of CRC, but also of biomarkers associated with different CRC stages or biomarkers relative to the response to treatment.

### 1.2. Colorectal Cancer, a Plethora of Classifications

Colorectal cancer is a very heterogeneous disease. Recent medical advances and the evolution of techniques have made it possible to provide anatomical, immunological, or mutational details for each stage of the disease. This increase in knowledge has led to the emergence of a number of colorectal cancer classifications to help manage the disease. Below are some examples of these different classifications.

In 2010, during the 7th edition of the American Joint Committee on Cancer, the various anatomic CRC classifications were clearly identified and compared (Table 1) [3]. The oldest classification is the Dukes classification (1932) [4], which classifies the various CRC stages from A to C. There is also the Astler-Coller system (1954), ranging from A to C3 [5]. Finally, the most recent classification (1988) is the TNM stage (for tumour, nodes, metastasis), where T represents the size of the tumour and its depth of penetration into the colonic wall (T1 to T4), N represents lymph node involvement (N0 to N2), and the M indicates the presence or absence of metastases (M0 or M1) [3].

CRC can also be classified according to its hereditary or non-hereditary nature, for example, sporadic or spontaneous cancer in the case of non-hereditary cancer, or familial or hereditary cancer linked to a genetic predisposition (30% of CRC) [6,7]. The best-known example of the latter is Lynch syndrome, also known as hereditary non-polyposis colorectal cancer (HNPCC). Lynch syndrome is an autosomal dominant disorder in which the *MLH1* (50%), *MSH2* (40%), *MSH6* (7–10%), and *PMS2* (5%) genes can be mutated (the percentages of these mutations being represented in parentheses). These genes are involved in DNA mismatch repair (MMR genes), and their mutations lead to accumulation of errors in DNA [8,9]. Familial adenomatous polyposis (FAP) is also an autosomal dominant disorder [10] involving mutations in the *APC* gene (adenomatous polyposis coli), a tumour suppressor gene and modulator of the Wnt/β-catenin pathway. Briefly, when *APC* is mutated, β-catenin is not degraded, thereby leading to a constitutive activation of the Wnt/β-catenin pathway and the transcription of the genes involved in cell cycle progression, cell migration, and the inhibition of cell adhesion, ultimately promoting the development of metastases [11,12]. Other less known or rare diseases include polyposis, associated with *MUTYH* or *NTHL1*, which are FAP forms affecting the *MUTYH* or *NTHL1* genes (both involved in DNA oxidative damage repair) [13,14,15], as well as Peutz–Jeghers syndrome and familial juvenile polyposis affecting the *LKB1*/*STK11* and *SMAD4* or *BMPR1A* tumour suppressor genes, respectively [16,17,18].

The type of carcinogenesis pathway is also a tool for classifying CRC. In the first category, CIN (chromosomal instability) tumours are characterised by chromosomal instabilities and represent 85% of CRC. They feature mutations which increase the probability of losing or gaining part of a chromosome or even an entire chromosome, the result of which can be an imbalance in the number of chromosomes per cell (aneuploidy), the latter being the main feature of CIN+ tumours. In CRC, the loss of chromosome 18q heterozygosity, containing the *SMAD4* and *DCC* genes, as well as the loss of other tumour suppressor genes, such as *APC*, *TP53* or *PTEN*, is most often observed. The best-known example of a CIN+ tumour is FAP [19]. A second category includes MSI+ (microsatellites instability) tumours, which exhibit microsatellite instabilities and account for 15% of CRC. These are hypermutated phenotypes caused by the loss of DNA mismatch repair activity [20]. In this group, one can distinguish hereditary MSI+, which represent 3% of MSI+ tumours, such as Lynch syndrome. The remaining 12% of MSI+ tumours are sporadic and involve hypermethylation of the *MLH1* promoter. These constitute a third class called CIMP+ (CpG island methylator phenotype) tumours, which are phenotypes with a hypermethylation of the genome, notably in tumour suppressor genes [21].

Many molecular classifications have also been inventoried with the aim of facilitating the clinical stratification of CRC and determining the best treatment option. These classifications are based on combinations of both clinical and histopathological parameters and gene characterisation [22]. The colon cancer subtype system (CCS) classifies three different subtypes: CCS1 (49%, *KRAS* and *TP53* mutations, high CIN, strong activation of Wnt signalling), CCS2 (24%, high MSI/CIMP, inflammatory cell infiltration, right colon), and CCS3 (27%, MSI/CIN, overexpression of EMT, matrix remodelling and cell migration genes, TGF-β pathway activation, *BRAF* and *PIK3CA* mutations) [23]. The colorectal cancer assigner system (CRCA) distinguishes five different classes in conjunction with the various cell types in the colonic crypt: Stem-like, goblet-like, transit-amplifying, inflammatory, and enterocyte [24]. The colon cancer molecular subtype system (CCMS) differentiates six different classes: C1 to C6 (21%, 19%, 13%, 10%, 27%, and 10%, respectively) [25]. The colorectal cancer intrinsic subtype system (CRCIS) distinguishes three different CRC subtypes: Type A (22%, MMR-deficient epithelial subtype), type B (62%, epithelial proliferative subtype), and type C (16%) [26,27]. Finally, the colorectal cancer subtyping consortium (CRCSC) distinguishes four CRC subtypes: CMS1 (14%, immune microsatellite instability), CMS2 (37%, canonical), CMS3 (13%, metabolic), and CMS4 (23%, mesenchymal) [28].

On the proteomic side, a study conducted by Zhang et al. analysed 95 samples of TCGA (the cancer genome atlas) colorectal tumors [29]. Using LC-MS/MS-based shotgun proteomic assays, they identified five proteomic subtypes for colorectal cancer using a consensus clustering method [30]. They then identified protein, gene, and clinical signatures, defining candidate biomarkers or therapeutic targets associated with each subtype. This work was one of the first significant studies combining proteomics and genomics on human samples, underlining the importance of proteogenomic integration to study CRC.

This non-exhaustive panel of CRC classifications highlights the considerable heterogeneity of CRC subtypes. These different classes have improved the management of CRC although, in many cases, the percentage of non-responders (NR) to treatment remains relatively high. One such example are patients with locally advanced rectal cancer (LARC). In North America, the standard treatment for these patients is neoadjuvant radiochemotherapy (NRCT) with 5-fluorouracil (5-FU); however, on average, 30% of these patients do not respond to treatment [31,32,33,34]. Efforts to facilitate the management of colorectal cancer and to avoid high levels of NR must therefore be maintained.

## 2. Proteomics for Biomarker Research

### 2.1. Biomarker Overview

According to the National Cancer Institute (NCI), a molecular biomarker is defined as a biological molecule found in blood, other body fluids, or tissues that is a sign of a normal or abnormal process, or of a condition or disease. The search for biomarkers, especially in cancer, is of high interest particularly for: (i) Detecting or assessing the risks of recurrence (diagnostic biomarkers), (ii) assessing the clinical prognosis, (iii) following the evolution of the disease or predicting a recurrence (prognostic biomarker), or iv) to improve personalised medicine by determining a patient’s response to a specific pharmacological treatment (predictive biomarker) [35]. For biomarkers associated with CRC, additional studies worth noting are from the studies of Jimenez et al. and De Wit et al., which list biomarkers, in general, identified from cell lines, murine models, and clinical material (tissue and serum) [36,37].

The development of biomarkers must follow a rigorous process in order to be used clinically. The first step in this process is pre-analytical validation. This step focuses on the samples that will be tested (time and storage conditions between sampling and processing, as well as after sample processing, time and duration of sample fixation, etc.) [38]. The second step consists of analytical validation. The analytical validity of a diagnostic test is its in vitro ability to perform the measurement of interest both accurately and reliably. In other words, does the test measure what it is intended to measure and correctly? During this step, many parameters relating to the test itself are recorded as accuracy (repeatability and reproducibility), sensitivity, specificity, linearity, limit of detection, and robustness of the test [38,39] (definitions of terms are provided in Table 2, top portion). Finally, clinical validation must provide evidence of the validity and clinical utility of the biomarker, again with many parameters to be measured and complied with, including clinical sensitivity, clinical specificity, probability of false-positives and false-negatives, relative risk, etc. [38,40] (definitions of terms provided in Table 2, bottom portion). Clinical validity corresponds to the ability of the test to accurately and reliably predict the clinical phenotype of interest. Clinical utility describes the ability to improve the clinical outcome of patients with measurable clinical events, and to provide added value in terms of optimisation of treatment decision and as a corollary of therapeutic strategy. Validating the predictive character of a marker is equivalent to demonstrating the clinical utility of its diagnostic test. As a result of these many, long and expensive steps, a large number of biomarkers have been described, albeit fail to reach the final stages to validate their clinical use [41].

### 2.2. Predictive Biomarkers and Proteomics

Predictive biomarkers, as well as other biomarker categories, can be of different types: Proteins, nucleic acids, antibodies, peptides, etc. [38] Many studies initially focused on gene mutation statuses, such that biomarkers and some of these statuses are already used to guide clinical decisions [43]. For example, in patients with advanced colorectal cancer, it has been shown that specific mutations in the *KRAS* gene render targeted therapies with cetuximab and panitumumab antibodies ineffective [44,45,46,47]. Indeed, these monoclonal antibodies target and inhibit the EGFR signalling pathway (cell growth and survival) [48]. In the presence of mutations mainly affecting codons 12 and 13, the *KRAS* gene becomes constitutively active, annihilating the antiproliferative effect of the treatment on cancer cells. Likewise, in patients with CRC, specific genotypes of the *UGT1A1* gene render the drug irinotecan ineffective [49,50]. Other similar predictive biomarkers also exist in other cancers, such as the detection of oestrogen and progesterone receptor expression to predict the response to hormone therapies in breast cancer [51], and the detection of HER-2 to predict the response to trastuzumab therapy, also in breast cancer [52,53].

However, in recent years, the search for protein biomarkers is becoming increasingly crucial. The primary goal of clinical proteomics is to study the proteome and its modifications in order to ascertain unique or signature biomarkers that can be used in clinical practice to promote personalised medicine [41]. Over the last ten years, approximately 4400 publications have been published regarding clinical proteomics based on the PubMed database.

The discovery of biomarkers by proteomics follows a series of distinct steps (Figure 1). This research first includes a so-called “discovery” or screening step owing to shotgun proteomics. This step can be carried out on small cohorts, as well as from patient samples (blood, fresh or fixed tumour biopsy, stools, etc.) whose protein content will be extracted and analysed by mass spectrometry. It consists of examining the proteome and the changes that occur in the latter to identify deregulated, overexpressed, or under-expressed proteins between different groups of patients (healthy or sick, responders or NR, etc.). Several techniques can be used to quantify proteins in samples [54], such as LFQ (semi-quantitative, widely used in proteomic shotgun studies), or labelling techniques, such as SILAC (stable isotope labelling by amino acids in cell culture) [55], TMT (tandem mass tag) [56], iTRAQ (isobaric tag for relative and absolute quantitation) [57], diLeu (dimethyl leucine) [58], or diART (deuterium isobaric amine reactive tag) [59]. The labelling techniques (SILAC, iTRAQ and other TMT, etc) allow for multiplexing of the samples and, thus, analyse several experimental conditions in a single injection in LC-MS/MS. There are two main advantages to this approach: Th reduction in experimental biases and in analysis time. The disadvantage of these techniques is either having to incorporate “weighted” amino acids into cells in culture, in the case of SILAC (3 conditions in maximum multiplexing), or to chemically bond “mass labels” to the peptides at the end of sample preparation. This is particularly the case with the iTRAQ or TMT-type reporter fragment techniques that allow up to 10 multiplexing conditions. For this reason, but also due to their relatively high cost, these labelling techniques are scarcely, or not at all, used in shotgun proteomics studies on fixed human samples in which LFQ is privileged. After the discovery step, the proteins with the largest changes between the various groups are selected for the verification step. This stage involves “targeted” proteomics using different techniques, such as SRM-MS (selected reaction monitoring-mass spectrometry), MRM-MS (multiple reaction monitoring-mass spectrometry), or SWATH (sequential window acquisition of all theoretical fragment ion spectra). This step will allow precise quantification of the target proteins. It generally involves a larger cohort than during the discovery stage, although the verification step is often performed on cellular or animal models before being applied to a cohort of patients. Finally, the validation step is the clinical assessment phase of the biomarkers.

For this latter step, very few studies have specifically been reported with regard to protein biomarkers for predicting treatment response from human samples and proteomics (Table 3). Katsila et al. investigated the secretome (secreted proteome) from plasma of 18 patients with metastatic colorectal cancer (mCRC), with the *KRAS* (exon 2) WT, and who were treated with chemotherapy and cetuximab. They noted that during cetuximab treatment, pEGFR levels increased in patients who responded well to treatment, did not evolve in patients with stabilisation of the disease, and decreased in patients with disease progression. The authors subsequently validated this hypothesis by measuring the secretome in 3D cultures obtained from several lines of SW48 colorectal cancer cells with different sensitivities to cetuximab [60]. This latter study is a good example of a proteomic study performed on human samples and validated by in vitro cell models, hence promoting the use of 3D cell cultures obtained directly from patient samples to pre-test the treatment. Martin et al. studied the serum of 23 mCRC patients prior to chemotherapy and bevacizumab. After two-dimensional difference gel electrophoresis (2D-DIGE) followed by LC-MS/MS, they identified 68 proteins differentially expressed between NR and responders. They selected three proteins for validation: Apolipoprotein E (APOE), vitamin D binding protein (DBP), and angiotensinogen (AGT), using ELISA and immunohistochemistry (IHC) [61]. These two validation techniques are those most used to validate biomarkers identified by proteomic discovery on patient samples, as they represent an easy and inexpensive means to perform these validations and often represent the tests that will be used in clinical laboratories. Croner et al. observed the proteome obtained from fresh preoperative tumour biopsies of 20 patients with LARC by ICPL (isotope coded protein label), followed by LC-MS/MS. They identified 140 differentially abundant proteins between responders and NR to NRCT with 5-FU ± oxaliplatin, four of which were validated by IHC: Plectin-1 (PLEC1), transketolase (TKT), trifunctional enzyme subunit mitochondrial precursor (HADHA), and transgelin (TAGLN) [62]. This latter instance represents an example of labelling prior to mass spectrometry analysis from fresh samples. Repetto et al. investigated the proteome obtained from preoperative frozen tumour biopsies of 15 patients with LARC by 2-DIGE, followed by LC-MS. They identified 30 differentially abundant proteins between responders and NR to NRCT with 5-FU ± oxaliplatin, and validated fibrinogen ß chain (FGB), three isoforms of actin, serpin B5 and B9, peroxiredoxin-4 (PRDX4), and cathepsin D (CTSD) by immunoblot. Finally, Chauvin et al. studied a proteome from formalin-fixed paraffin-embedded (FFPE) biopsies performed before NRCT with 5-FU in 23 LARC patients by proteomics shotgun (LC-MS/MS). They identified 384 differentially abundant proteins between NR and partial responders (PR), 248 between NR and total responders (TR), and 417 between PR and TR. They subsequently identified the proteins with the most significant differences between NR and TR, and identified six more abundant proteins in NR: Caldesmon (CALD1), mast cell carboxypeptidase 4 (CPA3), beta-1,3-galactosyltransferase 5 (B3GALT5), CD177 antigen (CD177), receptor-interacting serine/threonine-protein kinase 1 (RIPK1), and dihydropyrimidine dehydrogenase (DPYD) [63].

Altogether, the above findings feature a plethora of biomarkers identified in each study, very few of which are similar. The main impediment is that these studies involved small cohorts and are very different from one another. For example, in the latter three studies of Croner et al., Repetto et al., and Chauvin et al., despite a near-similar study for biomarkers predictive of response to RCTN with 5-FU in LARC patients, there were nonetheless many differences in experimental conditions: Storage of tumour biopsy (fresh [62], frozen [63], and FFPE [34]), classification of patients according to their response to treatment (Dworak [62], Mandard [63], AJCC [34]), as well as the use of different proteomics techniques (Table 3). With such different experimental conditions, it is difficult to achieve reproducibility.

The search for predictive biomarkers of the response to a given treatment can also be carried out using CRC cell lines (Table 4). Indeed, this model, although far from pathophysiological tumour conditions, represents an easy, inexpensive model to culture and to manipulate for obtaining resistant cell lines, and; therefore, constitutes a readily-usable first method for the search of potential biomarkers. Much of this type of study focuses on the comparison of a sensitive cell line versus the same resistant cell line [64,65,66]. It is indeed a strategy of choice for observing the differences between a sensitive model and a resistant model of similar origin, even if the process of induction of resistance in such a model modifies its genetic background. This latter approach can be used effectively for a proteomics discovery step to identify protein biomarker candidates responsible for resistance to a given treatment. It is also possible to further examine the mechanisms of drug resistance by implanting the cancer cells of interest into a xenograft model, such as the immunodeficient mouse [66,67]. These models represent a valuable method to study metastasis and genetic tumour evolution, although entail a relatively high cost and do not accurately reflect the pathophysiology of the tumour in the patient [68,69,70].

In an attempt to reproduce the pathophysiological tumour setting, a new study model has emerged over the past decade, namely the use of organoids. In 2009, Clevers’ team generated organoids from mouse jejunum via stem cell isolation from the intestinal crypt [73]. These organoids represent “mini-organs”, which self-organise from stem cells through the addition of different factors in the culture medium that reflect the tissue’s inherent characteristics [74]. This model was subsequently applied to a number of other tissues, including colorectal tumours [75], with many genetic manipulation techniques being adapted to this system (CRISPR-Cas9, shRNAs, siRNAs, retroviral infection, etc.) [76]. All of these characteristics, thus, position this model as a promising study strategy for personalised medicine by directly using the organoids derived from the patient’s tumour to test its response to a given treatment directly or via xenograft models (also known as patient-derived xenograft (PDX)) [77].

### 2.3. State-of-Art Tests Used in Clinical Practice

The vast majority of the tests currently in development or already available commercially are diagnostic tests (screening). Indeed, three stool-based tests are commonly available for the detection of pre-cancerous lesions: The guaiac-based faecal occult blood test (gFOBT), the faecal immunochemical test (FIT), and the Cologuard^TM^ test. The gFOBT allows the detection of blood in the stool through a chemical reaction enabled by the peroxidase activity of heme. This screening test is minimally invasive and inexpensive. The FIT is an improved version of the gFOBT, based on the reaction of an antibody with faecal haemoglobin. This test is preferred to the gFOBT as it can be performed without prior diet testing, unlike the gFOBT. The FIT has also been shown in many studies to be more sensitive and specific than the gFOBT (far fewer false-negatives and false-positives) [78,79,80,81,82]. Finally, the Cologuard^TM^ test was approved in August 2014 by the U.S. Food and Drug Administration (FDA). It measures occult haemoglobin and eleven DNA markers, including methylated vimentin, KRAS mutant, β-actin, methylated BMPR3, NDRG4 promoter regions, etc. (sensitivity = 93%, specificity = 87%) [83,84]. Other stool-based tests are currently being developed, such as the ColoSure^TM^ test, enabling the measurement of the methylation status of vimentin (sensitivity = 72–77%, specificity = 83–94%) [85]. The first and only blood test used for CRC screening was only recently approved in April 2016, being Epi proColon^TM^, and is based on blood detection of hypermethylation of the *SEPTIN9* gene. All of these tests are non-invasive and have been developed in order to overcome the problem of colonoscopy refusal [86,87]. Colonoscopy consists of introducing a probe (colonoscope) with a camera to detect precancerous or cancerous lesions. The major difficulty of this screening test is its invasiveness and the potential complications associated with its use [88]. Lastly, carcinoembryogenic antigen (CEA) represents another serum soluble protein used in clinical practice. CEA levels increase during CRC progression and decrease after surgery [89], although can also be elevated under certain conditions (smoking, ulcerative colitis, etc.) [90] making the latter a poor CRC screening biomarker. It is preferentially used to follow the evolution of the CRC and, in particular, as a prognostic biomarker of recurrence [91,92,93,94].

More recently, imaging mass spectrometry (IMS) has been gaining considerable momentum. This in situ and label-free technique allows the analysis of tissues in a very reliable and detailed way, in terms of localization and quantification of molecules such as proteins or peptides, lipids, cell metabolites, and drugs and their metabolites [95,96]. There are three different IMS techniques: MALDI [97], for peptide and protein detection; or desorption electrospray ionisation (DESI) [98] and secondary ion mass spectrometry (SIMS) [99], for smaller molecules. IMS represents a promising approach for the discovery and development of biomarkers. For example, Aichler et al. showed, by use of MALDI-IMS, that the clinical response to chemotherapy in patients with esophageal adenocarcinoma was related to mitochondrial defects [100]. Balluf et al. identified, also by us of MALDI-IMS, a seven-protein signature predicting outcomes in patients with gastric cancer after surgical resection [101].

## 3. Conclusions

Predictive biomarkers of response to treatment are highly valuable and the search for these biomarkers is a major public health challenge. Indeed, the discovery of a biomarker, or a signature of biomarkers, capable of predicting the response of patients with colorectal cancer to a given radiotherapy and/or chemotherapy treatment would invariably have many advantages: First of all, for the patient, in avoiding physically and psychologically demanding treatment; secondly, for physicians, in guiding their clinical decisions; and finally, for reducing health care costs (Figure 2) [34]. Unfortunately, despite numerous proteomic studies, no biomarkers identified after the discovery and/or validation steps are currently the subject of clinical studies [102,103], and this problem is very well discussed by Diamandis et al. [102]. However, as shown in Table 1 and Table 2, many proteins identified in the various proteomic studies are found in Zhang’s proteogenomic study (*) [29]. Indeed, Zhang et al. showed that the abundance of mRNA transcripts did not efficiently predict the abundance of proteins. This observation highlights the importance of proteogenomic integration to strengthen the data in the discovery of biomarkers, or of potential therapeutic targets in colorectal cancer [29,103]. Among the proteins emerging from different studies [34,61,62,63,64,65,66,67,71,72] and observed in Zhang’s proteogenomic study [29], we note the important presence of proteins involved in the regulation of the apoptotic process and in the negative regulation of the cellular protein metabolic process (Figure 3), identifying common proteins and possible biomarkers that overlap between these studies. It would; therefore, be interesting to encourage efforts towards these pathways.

It is also important to remember the importance of targeted proteomics to validate identified biomarkers after proteomic shotgun studies, for example; all this in order to strengthen the study and open the way for the clinical validation step. Finally, the development of IMS is becoming increasingly important in the search for biomarkers, and represents a promising tool for progress in the fight against cancer.

For the time being, clinical studies currently focus on the development of screening tests with the aim of making these tests as minimally invasive as possible (blood or stool collection), and primarily target genes and not proteins. Efforts to discover protein biomarkers predictive of treatment response must, nevertheless, continue in order to improve the quality of life of these patients, whilst offering them the best treatments and; therefore, a better chance of survival.

## Figures and Tables

**Figure 1 proteomes-06-00049-f001:**
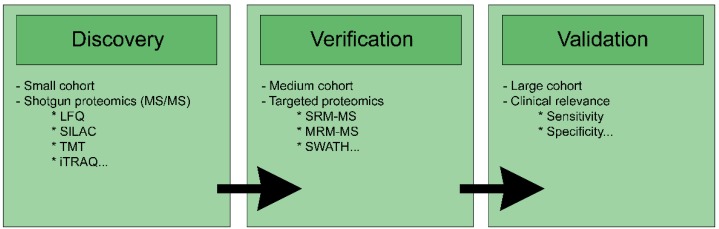
Steps for biomarker discovery by proteomics. MS: Mass spectrometry; LFQ: Label free quantification; SILAC: Stable isotope labelling by amino acids in cell culture; TMT: Tandem mass tag; iTRAQ: Isobaric tag for relative and absolute quantitation; SRM: Selected reaction monitoring; MRM: Multiple reaction monitoring; SWATH: Sequential window acquisition of all theoretical fragment ion spectra.

**Figure 2 proteomes-06-00049-f002:**
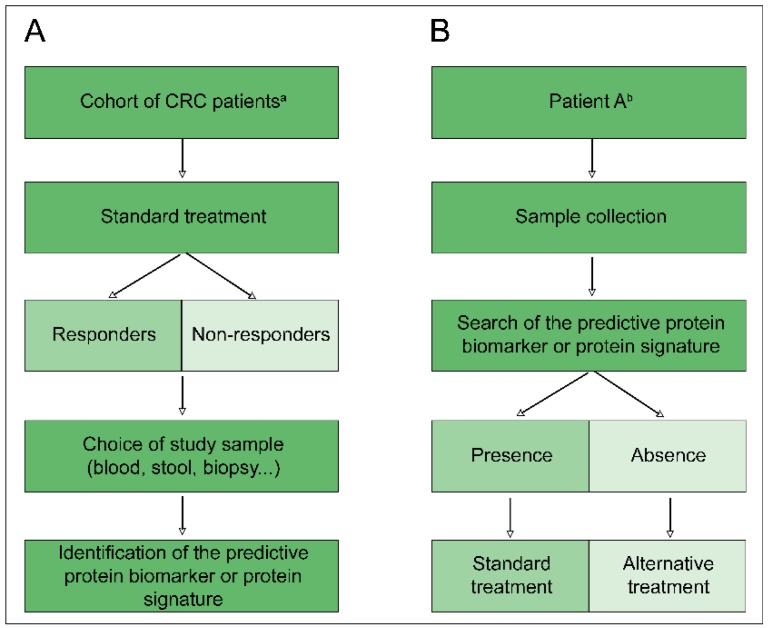
Ideal course of protein biomarker research predictive of treatment response. (**A**) Screening step of a patient cohort to determine a predictive protein biomarker or predictive protein signature. (**B**) Clinical application of a predictive protein biomarker or predictive protein signature for personalisation of treatment according to the patient. CRC: Colorectal cancer; a: Key features for determining the inclusion parameters of a patient in the cohort, according to a classification detailed in the introduction; b: Patient meeting all of the inclusion criteria defined during the screening step.

**Figure 3 proteomes-06-00049-f003:**
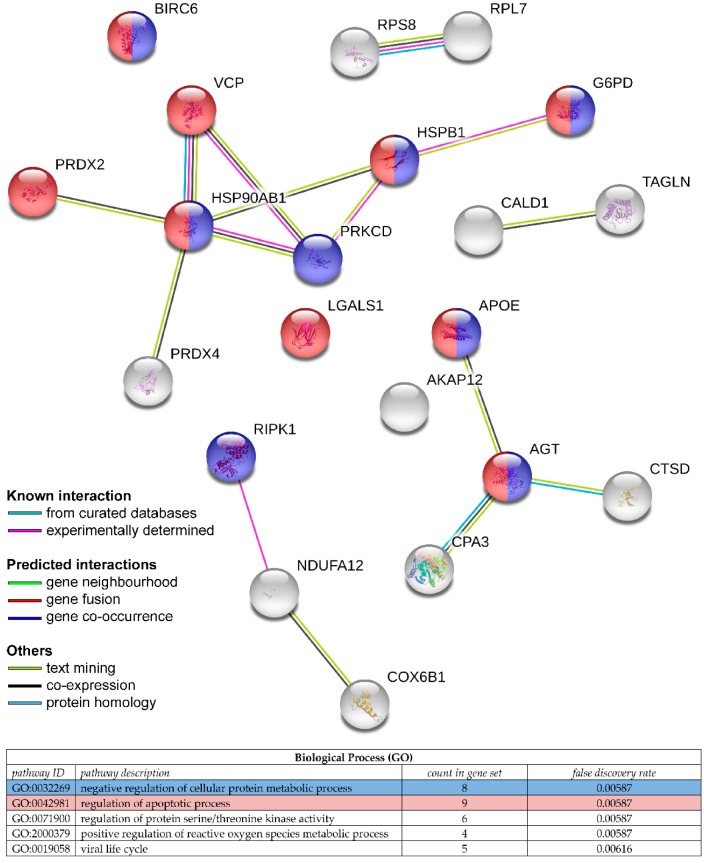
Representation by STRING software of potential protein biomarkers identified in different proteomic studies (https://string-db.org/). The nature of the interactions is shown at the bottom left of the figure and lists the known interactions. Biological processes are annotated in the table at the bottom of the figure.

**Table 1 proteomes-06-00049-t001:** Colon and rectum cancer staging [3].

Anatomic Stage					
Stage	T	N	M	Duke’s	MAC
0	Tis	N0	M0	-	-
I	T1, T2	N0	M0	A	A B1
IIA	T3	N0	M0	B	B2
IIB	T4a	N0	M0	B	B2
IIC	T4b	N0	M0	B	B3
IIIA	T1–T2, T1	N1/Nc N2a	M0	C	C1
IIIB	T3–T4a, T2–T3, T1–T2	N1/N1c, N2a, N2b	M0	C	C2, C1/C2, C1
IIIC	T4a, T3–T4a, T4b	N2a, N2b, N1–N2	M0	C	C2, C2, C3
IVA	Any T	Any N	M1a	-	-
IVB	Any T	Any N	M1b	-	-

T: Tumour; N: Lymph nodes; M: Metastasis; MAC: Modified Astler-Coller classification.

**Table 2 proteomes-06-00049-t002:** Definitions of common terms used in the analytical validation and clinical validation steps.

Terms	Definitions
**Analytical Validation**	
Accuracy	Agreement between a test result of a quantity and its reference value
Repeatability	Describes test results performed under the same conditions
Reproducibility	Describes test results performed under different conditions
Analytical Sensitivity	The ability of the assay to obtain a concordance in positive results between assay and reference method
Analytical Specificity	The ability of the assay to obtain a concordance in negative results between assay and reference method
Linearity	The ability of the assay to yield a proportional effect between test values and concentrations of the analyte in the sample
Limit of Detection	The lowest concentration of analyte significantly different from zero or negative control
Robustness	Test precision following deliberate changes in assay conditions (temperature, storage, etc.)
**Clinical Validation**	
Clinical Sensitivity	Ability of a biomarker to predict a change in a clinical endpoint (relationship between the magnitude of change in the biomarker and the magnitude of change in the clinical endpoint)
Clinical Specificity	Ability of a biomarker to distinguish responders and NR patients in terms of changes in clinical endpoints
Relative Risk	Ratio of the probability of an event (e.g., disease recurrence, death) occurring in the treated group to the probability of the event occurring in the control group

Sources: Jennings et al., 2009 [42] and Masucci et al., 2016 [39] (analytical validation); and Dobbin et al., 2016 [40] (clinical validation). NR: Non-responder.

**Table 3 proteomes-06-00049-t003:** Protein biomarkers predictive of treatment response in colorectal cancer obtained by proteomics from human material.

Biological Sample Type	Proteomic Approach	Treatment	Identified Candidate Biomarkers	Reference
Secretome	LC-MS/MS	Cetuximab + FOLFIRI	Phospho-epidermal growth factor receptor (p*EGFR*) [P00533]	[60]
Serum	2D-DIGE + LC-MS/MS	Bevacizumab + XELOX or FOLFOX	Apolipoprotein E (*APOE*) [P02649] *, angiotensinogen (*AGT*) [P01019] *, D site-binding protein (*DBP*) [Q10586]	[61]
Tumour biopsy	ICPL + LC-MS/MS	NRCT 5-FU/capecitabine ± oxaliplatin	Plectin (*PLEC1*) [Q15149], transketolase (*TKT*) [P29401], trifunctional enzyme subunit alpha, mitochondrial (*HADHA*) [P40939], transgelin-2 (*TAGLN*) [P37802] *	[62]
Tumour biopsy	2-DIGE + LC-MS	NRCT 5-FU/capecitabine ± oxaliplatin	Fibrinogen ß chain (*FGB*) [P02675], actin (three isoforms), serpin B5 (*SERPINB5*) [P36952], serpin B9 (*SERPINB9*) [P50453], peroxiredoxin-4 (*PRDX4*) [Q13162] *, cathepsin D (*CTSD*) [P07339] *	[63]
Tumour biopsy	LC-MS/MS	NRCT 5-FU/capecitabine	Caldesmon (*CALD1*) [Q05682] *, mast cell carboxypeptidase 4 (*CPA3*) [P15088] *, beta-1,3-galactosyltransferase 5 (*B3GALT5*) [Q9Y2C3], CD177 antigen (*CD177*) [Q8N6Q3], receptor-interacting serine/threonine-protein kinase 1 (*RIPK1*) [Q13546] *, dihydropyrimidine dehydrogenase (*DPYD*) [Q12882], NDUF proteins (complex 1 of the mitochondrial respiratory chain) *, ribosomal proteins (small/large subunits) *	[34]

LC-MS/MS: Liquid chromatography tandem-mass spectrometry; 2D-DIGE: Two-dimensional difference gel electrophoresis; FOLFIRI: 5-fluorouracil and irinotecan; XELOX: Capécitabine (Xeloda^®^) and oxaliplatin; FOLFOX: 5-fluorouracil and oxaliplatin; ICPL: Isotope-coded protein label; NRCT: Neoadjuvant radio-chemotherapy; 5-FU: 5-fluorouracil. (*) Proteins identified in Zhang’s proteogenomic study [29].

**Table 4 proteomes-06-00049-t004:** Protein biomarkers predictive of treatment response in colorectal cancer obtained by proteomics from colorectal cancer cell lines or animal models.

CRC Cell Line	Proteomic Approach	Study Focus	Identified Candidate Biomarkers	Reference
HCT-116	iTRAQ, ICAT; LC MALDI-TOF/TOF MS	Butyrate response	Heat shock protein HSP 90-β (*HSP90AB1*) [P08238] *, galectin-1 (*LGALS1*) [P09382] *, A-kinase anchor protein 12 (*AKAP12*) [Q02952] *, vesicle-trafficking protein SEC22b (*SEC22B*) [O75396] *, cytochrome c oxidase 6b1 (*COX6B1*) [P14854] *	[71]
SW620	LC MALDI-Q-TOF MS/MS	Irinotecan resistance	α-enolase (*ENO1*) [P06733], cofilin (*CFL1*) [P23528], peroxiredoxin-2 (*PRDX2*) [P32119] *	[64]
Colonospheres derived from liver metastases	LC-MS/MS	Cisplatin and oxaliplatin resistance	Baculoviral IAP repeat-containing protein 6 (*BIRC6*) [Q9NR09] *	[72]
DLD-1	2-DIGE; LC MALDI-TOF/TOF MS	5-FU resistance	Heat shock protein beta-1 (*HSPB1*) [P04792] *, proteasome subunit α type-5 (*PSMA5*) [P28066], transitional endoplasmic reticulum, ATPase (*VCP*) [P55072] *, 14-3-3 protein β (*YWHAB*) [P31946], 14-3-3 protein γ (*YWHAG*) [P61981], 14-3-3 protein σ (*SFN*) [P31947], phosphoglycerate kinase 1 (*PGK1*) [P00558]	[65]
GEO	2-DIGE; LC-MS	Cetuximab resistance	Glucose-6-phosphate 1-dehydrogenase (*G6PD*) [P11413] *, L-lactate dehydrogenase B chain (*LDHB*) [P07195], pyruvate dehydrogenase E1 component subunit alpha, somatic form, mitochondrial (*PDHA1*) [P08559], transketolase (*TKT*) [P29401]	[66]
HCT-116	LC-MS/MS	Dasatinib (Src-selective inhibitor) resistance	pY313-protein kinase C delta type (*PRKCD*) [Q05655] *	[67]

iTRAQ: Isobaric tag for relative and absolute quantitation; ICAT: Isotope-coded affinity tag; LC-MS/MS: Liquid chromatography tandem-mass spectrometry; 5-FU: 5-fluorouracil; LC-MS: Liquid chromatography-mass spectrometry; MALDI-TOF MS: Matrix assisted laser desorption ionization-time of flight mass spectrometry; MALDI-Q-TOF: MALDI-quadrupole-time of flight; 2D-DIGE: Two-dimensional difference gel electrophoresis. (*) Proteins identified in Zhang’s proteogenomic study [29].

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
