# Peer review of "Clinical Proteomics in Colorectal Cancer, a Promising Tool for Improving Personalised Medicine"

_proteomes, 2018, doi:10.3390/proteomes6040049_

Round 1

Reviewer 1 Report

In the submitted review entitled “Clinical Proteomics in Colorectal Cancer, a Promising Tool for Improving Personalised Medicine”, the authors reviewed the different biomarkers discovered through proteomics studies in various colorectal cancer study models (blood, stool, biopsies) including the different proteomic techniques used for the discovery of these biomarkers, as well as the various tests used in clinical practice and those currently in clinical phase. They also discussed the limits and perspectives related to proteomic biomarker research for personalised medicine in colorectal cancer.  Overall, the review has a clear structure and is clearly written. However, some important references on biomarker discovery and application in colorectal cancer, and MS strategies are missing, which should be addressed in the revised version. Here are some examples for the references on biomarker discovery and application in colorectal cancer:

1.      The Nature paper of the TCGA CRC tumors of the Liebler lab: Zhang B et al., Proteogenomic characterization of human colon and rectal cancer. Nature. 2014 Sep 18;513(7518):382-7)

2.      Ma H, et al. Mass spectrometry (MS)-based translational proteomics for biomarker discovery and application in colorectal cancer, Proteomics Clin Appl., 2016,10, 503-515.

       3.  Jimenez, C. R., et al., Proteomics of colorectal cancer: overview of discovery

studies and identification of commonly identified cancerassociated proteins and candidate CRC serum markers. J. Proteomics 2010, 73, 1873–1895.

        4.  De Wit, M., et al., Proteomics in colorectal cancer translational research:

biomarker discovery for clinical applications. Clin. Biochem. 2013, 46, 466–479.

  5.  De Wit M, et al., Cell surface proteomics identifies glucose transporter type 1 and prion protein as candidate biomarkers for colorectal adenoma-to-carcinoma progression. Gut. 2012; 61(6):855-64

In addition, please also add some discussions on imaging mass spectrometry (IMS), which can be used to measure the spatial arrangement and relative concentration of biomarkers in biological samples, from elements to large proteins. Meanwhile, developed imaging technology can utilize MALDI and TOF-MS to generate profiles and two-dimensional ion density maps of peptide and protein signals directly from the surface of thin tissue sections and this approach is ideal for biomarker detections..

Author Response

Responses to Reviewer 1

We are pleased to present you a revised version of our manuscript. We would like to thank the reviewers for their time and their useful comments. We hope that the additional information and references that we present in this revised manuscript now clarify and add substantial new information that will satisfy the reviewers.

Please find below are point by point answers to your suggestions:

However, some important references on biomarker discovery and application in colorectal cancer, and MS strategies are missing, which should be addressed in the revised version. Here are some examples for the references on biomarker discovery and application in colorectal cancer:

1.    The Nature paper of the TCGA CRC tumors of the Liebler lab: Zhang B et al., Proteogenomic characterization of human colon and rectal cancer. Nature. 2014 Sep 18;513(7518):382-7.

Response: We now discuss Zhang's publication at the end of part 1.2 (references 29 and 30) and in the conclusion. We also demonstrated the proteins identified in Zhang's proteogenomic study in Tables 1 and 2.

2.    Ma H, et al. Mass spectrometry (MS)-based translational proteomics for biomarker discovery and application in colorectal cancer, Proteomics Clin Appl., 2016,10, 503-515.

Response: We added the reference from Ma et al. in the conclusion (reference 103) where appropriate.

3.    Jimenez, C. R., et al., Proteomics of colorectal cancer: overview of discovery studies and identification of commonly identified cancer-associated proteins and candidate CRC serum markers. J. Proteomics 2010, 73, 1873–1895.

Response: We added Jimenez’s review in the part 2.1 (reference 36) as appropriate and incorporated in the text.

4.    De Wit, M., et al., Proteomics in colorectal cancer translational research: biomarker discovery for clinical applications. Clin. Biochem. 2013, 46, 466–479.

Response: We are now referring to De Wit’s review in the part 2.1 (reference 37).

5.    De Wit M, et al., Cell surface proteomics identifies glucose transporter type 1 and prion protein as candidate biomarkers for colorectal adenoma-to-carcinoma progression. Gut. 2012; 61(6):855-64.

Response: We decided not to mention this reference since our review focuses mainly on predictive biomarkers of response to treatment, which is already a very large subject as exemplified by our over 100 references in such a short review. This article relates to predictive biomarkers of adenoma-to-carcinoma progression, which is not really the subject of this review.

In addition, please also add some discussions on imaging mass spectrometry (IMS), which can be used to measure the spatial arrangement and relative concentration of biomarkers in biological samples, from elements to large proteins. Meanwhile, developed imaging technology can utilize MALDI and TOF-MS to generate profiles and two-dimensional ion density maps of peptide and protein signals directly from the surface of thin tissue sections and this approach is ideal for biomarker detection.

Response: Indeed, this is a very interesting topic in itself. We added a paragraph describing IMS at the end of section 2.3. We hope the special issue will have a review dedicated on this important subject in clinical proteomics.

Reviewer 2 Report

In this manuscript, Chauvin and Boisvert provide a succinct (compared to the vastness of the topic covered) review on the biomarkers that have been discovered through proteomics studies in various colorectal cancer (CRC) studies.

This review includes 1) a short description of CRC classification; 2) a description of proteomic techniques used in biomarker discovery; 3) a scheme of the steps required from discovery to validation phases; 4) list of biomarkers organised according to the type of biological sample or cell model and pharmacological treatment.

Overall, this review is well written and meets the interest to the broad readership of Proteomes.

However, after reading the manuscript, I was left with a taste of dissatisfaction. I expected the authors would have concluded with a list of proteins to suggest as a molecular signature useful to diagnose/prognose CRC. Instead, they conclude writing “unfortunately, despite numerous proteomic studies, no biomarkers identified after the discovery and/or validation steps are currently the subject of clinical studies.” In fact, according to the literature cited by the authors, each model/sample type/proteomic approach resulted with different lists of biomarkers, without any overlap.

I think this review do not add much compared to other reviews already published on this topic (see, for example, Lee PY 2018, https://doi.org/10.1016/j.jprot.2018.06.014;  Alvarez-Chaver 2014, doi:10.3748/wjg.v20.i14.3804).

Author Response

Responses to Reviewer 2

We are pleased to present you a revised version of our manuscript. We would like to thank the reviewers for their time and their useful comments. We hope that the additional information and references that we present in this revised manuscript now clarify and add substantial new information that will satisfy the reviewers.

Please find below are point by point answers to your suggestions:

In this manuscript, Chauvin and Boisvert provide a succinct (compared to the vastness of the topic covered) review on the biomarkers that have been discovered through proteomics studies in various colorectal cancer (CRC) studies.

Response: We wanted to focus on predictive biomarkers of the response to treatment in humans since there are already many reviews listing prognostic and diagnostic biomarkers, in cell lines or in murine models. We completely agree that this is a very large topic, and thus wanted to focus on these biomarkers, which already include over 100 references.

However, after reading the manuscript, I was left with a taste of dissatisfaction. I expected the authors would have concluded with a list of proteins to suggest as a molecular signature useful to diagnose/prognose CRC. Instead, they conclude writing “unfortunately, despite numerous proteomic studies, no biomarkers identified after the discovery and/or validation steps are currently the subject of clinical studies.” In fact, according to the literature cited by the authors, each model/sample type/proteomic approach resulted with different lists of biomarkers, without any overlap.

Response: You are right that in the original submitted version of the review, we did not committed ourselves in proposing specific proteins that stand out from analysing these studies. To answer this criticism and bring our analysis further, we added a link between the proteins discovered in the studies listed in Tables 1 and 2 and the proteogenomic study of Zhang et al., and came up with a list of proteins that can be found in several studies, and thus present an overlap. These proteins are now marked with an asterisk (*) in both tables.  We analysed these overlapping proteins and found two different pathways there were found enriched in function (metabolic and apoptosis), which highlights proteins that are potentially interesting biomarkers that were identified in several of the proteomics studies. This was incorporated in the text, and we added Figure 3 showing a pathway analysis of proteins found in several of the proteomics studies.

Round 2

Reviewer 1 Report

Since the authors have revised to address all the comments raised by the reviewer, and significant improvement has been made. Acceptance of the work is thus suggested.